# Unveiling the Chaos in Postural Control in Adults with Achondroplasia

**DOI:** 10.3390/jfmk9010039

**Published:** 2024-02-26

**Authors:** Inês Alves, Maria António Castro, Sofia Tavares, Orlando Fernandes

**Affiliations:** 1School of Health and Human Development, Évora University, 7004-516 Évora, Portugal; orlandoj@uevora.pt; 2Comprehensive Health Research Centre (CHRC), Évora University, 7004-516 Évora, Portugal; 3ANDO Portugal, National Association of Skeletal Dysplasias, 7005-144 Evora, Portugal; 4School of Health Sciences, Polytechnic Institute of Leiria, 2411-901 Leiria, Portugal; maria.castro@ipleiria.pt; 5RoboCorp Laboratory, i2a–IPC, CEMMPRE, University of Coimbra, 3004-531 Coimbra, Portugal; 6CIEP UÉ, Department of Psychology, Évora University, 7004-516 Evora, Portugal; tavares.sofia@uevora.pt

**Keywords:** center of pressure, nonlinear measures, limb lengthening, strength, physical activity

## Abstract

Background: Achondroplasia is a rare genetic skeletal condition characterized by disproportionate short stature. There is limited evidence on postural control in adults with achondroplasia and how lower limb lengthening (intervention) interacts with body dynamics. This study investigated sway variability during quiet standing in adults with achondroplasia with natural growth (N) and with lower limb lengthening (LL). Methods: Sixteen adults performed bilateral/unilateral standing tasks. Linear (total excursion, amplitude, and ellipse area) and nonlinear (sample entropy and correlation dimension) center of pressure sway metrics were analyzed in the anteroposterior/mediolateral directions. Relationships between posture metrics, strength, and physical activity were explored. Between-groups statistics were calculated. Results: The LL group exhibited amplified linear sway, indicating larger postural deviations, and reduced sample entropy and correlation dimension, indicative of more rigid and repeated corrections. The N group exhibited more unpredictable and adaptive movement corrections. Numerous correlations emerged between strength and posture measures, with relationships altered by intervention. Conclusions: Adults with achondroplasia display distinct balance strategies influenced by intervention. The results indicate that LL is associated with altered variability and adaptability compared to natural development. Relationships with muscle strength spotlight a key role of muscle capacity in postural control modulation after growth alterations in this population.

## 1. Introduction

Skeletal dysplasias (SD) are a heterogenous group of 771 rare bone conditions [1] with short stature as a common feature. Achondroplasia (ACH) is one of the best-known SD, with a prevalence of 1:25,000 births. People with achondroplasia (PwACH) present disproportionate short stature due to proximal shortening of the upper and lower limbs [2] as the result of a gain-of-function mutation of the fibroblast growth factor receptor type 3 (*FGFR3*) gene, heavily impairing longitudinal bone growth [2]. Adult height is on average 132 cm in men and 124 cm in women, resulting in a total average height deficit of −6.0 standard deviation scores in both genders [3]. This population also has strong predisposition to abdominal obesity [4].

To increase height, a surgical procedure defined by lower limb lengthening (intervention) has been an option for PwACH since the early 20th century [5]. The prevalence of limb lengthening varies from 1.2% in a study from the USA [6] to 21.5% in a study in Europe [7], with higher prevalence in Spain and Italy. The intervention functions based on external fixators and, more recently, intramedullary nails [8] to distract long bones, allowing new bone formation in between. Some PwACH choose to undergo lengthening on the tibias, femurs, or both, adding to their final height by up to 15 to 40 cm [9], with lengthening of 50% over initial bone length considered extensive [10]. Limb lengthening is a method based on the tension-stress principle [10]. Gradual traction creates stresses that can stimulate and maintain the regeneration and active growth of certain tissues such as bone. With adequate blood supply, steady gradual traction of the tissues activates proliferative and biosynthetic functions [10]. Previous studies identified sagittal imbalance to be significantly different in children with ACH compared to controls, with nearly 50% presenting thoracic hypokyphosis and more than half presenting an increase in lumbar lordosis with pelvic incidence, as well as the angle of the pelvic tilt and vertical body axis [11,12,13]. In relation to lower limb lengthening, there was an observed increase in the lumbosacral angle up to normal values in all age groups in the postsurgical period [12], with increased mobility of the femoral heads, which led to changes in pelvic parameters, a decrease in lumbar lordosis, an increase in thoracic kyphosis, and correction of the sagittal body imbalance [12]. However, there have been no longitudinal studies assessing the clinical outcome of balance and postural control of lower limb lengthening in adults with achondroplasia. Human posture is regulated via several feedback loops to adjust for both internal and external changes [14]. Posture is defined as the geometric relation between two or more body segments, and this relation is expressed in terms of joint angles between segments [15]. Variability is a natural and important feature of human movement. Mature motor skills and healthy states are associated with an optimal amount of movement variability. This variability also has form and is characterized by a chaotic structure. Less-than-optimal movement variability characterizes biological systems that are overly rigid and unchanging, whereas greater-than-optimal variability characterizes systems that are noisy and unstable. Both situations characterize systems that are less adaptable to perturbations, such as those associated with abnormal motor development or unhealthy states [16]. The standard for the measurement of postural control is posturography, which quantifies body sway using center of pressure (CoP) data recorded by a force plate [17]. Postural sway in a standing position is a process of continuous small body deviations to maintain static balance in the orthostatic position, and it is an indicator of the quality and characteristics of posture regulation [14]. Using a sensorimotor control system that measures body segment displacement, muscle activity, and the motion patterns of the center of mass (CoM) and CoP, it is possible to conduct an evaluation of postural control [15].

It seems necessary to collect existing data related to the use of chaos indicators to assess postural control [18]. CoP represents the dynamics of the neuromusculoskeletal system, which can be scaled in time and space with sources of nonlinearities as well as sensory–motor redundancies to control motion in each single axis. Traditional biomechanical models of postural stability do not fully characterize the nonlinear properties of postural control [19]. The inherent nonlinearities in the human postural control system, such as the complex sway pattern, are an integral part of the postural dynamics [20]. Nonlinear measures can capture the temporal component of the variation in CoP displacement regarding how motor behavior develops over time. Therefore, these measures make it possible to quantify regularity, adaptability to the environment, stability, and complexity [18]. While studies on postural control have focused mostly on linear measures of the CoP trajectories, increased use of nonlinear methods has been progressively applied to determine the dynamics of the CoP trajectories in different populations [21].

The maintenance of balance and body orientation in the standing position is essential for performing activities of daily life as well as for practicing physical and sports activities; the most common assessment tool to quantify postural balance is a static standing test with eyes open or closed [18]. Additionally, maintaining an upright stance requires controlling the inherently unstable multi-joint human body within a small base of support, despite the biological motor and sensory noise that challenges balance [22]. This base of support is even smaller for PwACH who undergo limb lengthening.

Postural balance depends on body size and muscle tone as well as the visual, vestibular, somatosensory, and central nervous systems [23,24]. Previous studies in the general population have identified altered integration of somatosensory inputs in sedentary men with a high total fat mass percentage. These factors likely affect the tuning, sequencing, and execution of balance strategies [25]. Linear measures such as CoP total excursion (ToTex), amplitude (A) and ellipse area (Area) quantify the amount of CoP movement during a specific task, independently of their order in the distribution. The nonlinear system approach helps to evaluate different aspects of the CoP data, and nonlinear measures make it possible to quantify the regularity and complexity of the system [26]. Nevertheless, nonlinear measures can capture the temporal component of the variation in CoP displacement regarding how motor behavior develops over time [18] and can provide indirect insight into the functioning of the nervous system [26]. The nonlinear measures selected for this study included sample entropy (SaEn), which indexed unpredictability in the fluctuating trajectory. This measure can be used to estimate the randomness of a series of data without any previous knowledge about the source generating the dataset [27]. Sample entropy is not a measure of complexity but provides information on an aspect of complexity, namely, how regular or predictable the data are, in which higher entropy means lower predictability [28]. Additionally, the correlation dimension (CoDim) measures the dimensionality of the time series in place, with higher dimensions pointing to more complex spatial corrections. It can detect subtle changes in postural sway patterns and allow characterization of the dimensionality of posture dynamics in shorter datasets, as in this study. Previous studies in the general adult population have highlighted the existence of linear relations between physical activity, handgrip strength, body composition, and multimorbidity [29]. It has been reported that PwACH present poor physical fitness, low participation in physical activities and reduced energy expenditure compared with the general population [30,31,32,33]. While muscle size is a predetermining factor of muscle strength, PwACH present shorter stature, smaller muscle size and lower maximal voluntary contraction strength than average-height people [34]. Additionally, there may exist a center-of-gravity displacement in PwACH [35], although there are no confirmations of this point.

Studying populations with rare conditions, such as PwACH, is of paramount importance for a multitude of reasons from both a medical and a societal perspective. This significance can be evidenced through various practical aspects such as advancing our understanding of disease mechanisms, enhancing drug development, and improving personalized care and support systems [36].

The aim of this study was to characterize postural control during quiet standing, infer differences between adults with achondroplasia (AwACH) with natural growth development and those who underwent lower limb lengthening, and infer whether muscle strength and physical activity (PA) are associated with posture and balance.

## 2. Materials and Methods

### 2.1. Data Collection

Sixteen AwACH, comprising 10 women and 6 men (37.2 ± 13.6 years), volunteered to participate in this cross-sectional study, registered in the Open Science Framework repository at https://osf.io/ype4a (accessed on 22 December 2023). Data collection allowed two groups to be identified: participants with natural growth (N) and those with previous lower lengthening surgery (LL). The time elapsed after limb lengthening surgery in the LL group varied between 5 to 32 years, and the total gain in height varied between 12 and 34 cm. All participants signed an informed consent previously approved by the Ethics Committee of the researchers’ university and by the advocacy organization. The inclusion criteria for the study were to be over 18 years old and have achondroplasia. The exclusion criteria were to have vision loss/alteration, vestibular impairments, neuropathy, or an inability to stand upright unsupported. Evaluations were conducted between November 2022 and March 2023. All measurements were collected on the same day for each participant. Postural acquisitions of CoP were collected with a Bertec^®^ force plate, model FP4060-07-1000 (Bertec Corporation, Columbus, OH, USA), at a frequency of 1000 Hz. Data collection was conducted in a sequential process during a 30-s period in each of the following tasks: bipedal standing with eyes open (O), bipedal standing with eyes closed (C), and unipedal standing on the right foot (R) and left foot (L) with eyes open, with a 60 s interval between each task’s data acquisition. The data were processed using a custom MATLAB code (R2018a; Mathworks, Inc.; Natick, MA, USA), with frequency downsampling to 100 Hz, which allowed calculation of the traditional and nonlinear parameters normally used for postural control, including total excursion (ToTex), ellipse area (Area), and amplitude (A). Anteroposterior (AP) and mediolateral (ML) directional components were extracted separately. For nonlinear measures, time series were analyzed for sample entropy (SaEn) and correlation dimension (CoDim). Visual inspection was used to exclude transitional movements before steady-state epochs, as these do not translate static postural control behavior. The data were processed with a low-pass filter at 15 Hz, using a 4th-order Butterworth filter to reduce unwanted noise components. Selecting the right cut-off frequency for a digital filter is essential for the accuracy and reliability of data interpretation, particularly in studies involving the analysis of oscillations in the center of pressure. This optimal frequency varies based on several factors. These include the characteristics of the study population, the conditions of data collection, and the specific aims of the research, as highlighted by Prieto et al. (1996) and Duarte and Freitas (2010) [24,37]. Cut-offs were selected after conducting a pilot study and analyzing the signal characteristics from the force platform, as suggested by Cornwall and McPoil (1994), along with considering the insights of Winter (1995) [38,39]. This balance is critical to ensure that vital information related to postural control is not lost, especially when conducting nonlinear analyses. The goal is to retain the integrity of the data while effectively filtering out irrelevant noise, thereby ensuring the validity of the study’s conclusion. Nonlinear analysis first included a surrogation test on each trial to verify that nonlinear methods were appropriate to use and that the time series data were different from those of a randomly generated time series. Subsequently, the sample entropy (according to Hernandez, 2016) of the CoP was calculated in the ML (SaEn_ML) and AP (SaEn_AP) directions [40].

For physical assessments, body measurements of height and foot length (FL) were collected with an anthropometric kit (GPM Large Instrument Kit 113, Seritex, East Rutherford, NJ, USA), and body composition for fat mass percentage (FM) was estimated with a bioimpedance scale (Tanita MC780-PMA), as was weight. Participants were also requested to perform the following physical evaluations: maximum number of regular push-ups (PU) during 30 s and the 6 min walking test (6MWT), performed indoors along a straight corridor over a length of 30 m [41]. Handgrip strength (HGS) was measured with a Jamar Plus+ Digital Hand Dynamometer, 200Ib (Sammons Preston Rolyan, Bolingbrook, IL, USA). Participants had two trials recorded for the dominant hand (right hand for all participants), and the average value of the two attempts was recorded. To identify the type and level of physical activity in the last 7 days prior to postural acquisitions, the study applied a self-reported questionnaire, the International Physical Activity Questionnaire (IPAQ), short version [25]. Data collected with the IPAQ consist of the median value of the physical activity score (PAS) presented as the combined total physical activity MET-min/week, computed as the sum of Walking + Moderate + Vigorous MET-min/week scores. Physical activity levels (PAL) from the IPAQ are defined as 1. inactive, 2. minimally active, and 3. HEPA active, in ascending level of physical activity.

### 2.2. Statistical Analysis

Sample size was calculated based on power analysis using G*Power software (Franz Faul, Edgar Erdfelder, Axel Buchner, Universität Kiel, Kiel, Germany, version 3.1.9.6) [23]. With α = 0.05 and a power of 0.90, a minimum sample size of 15 individuals was needed to achieve a large effect size (d = 0.8). Statistical analyses were performed using the Jamovi statistics platform, The jamovi project (2023), version 2.3, computer Software, Australia. As the sample was small, the normality of the data distribution was assessed using the Shapiro–Wilk test for all measures from each group (N and LL). As for anthropometric variables, the assumption of a normal distribution was rejected (*p* < 0.05) for 25 measure combinations and accepted (*p* > 0.05) for 27, either in bipedal stance, eyes open or closed, or in unipedal stance on the right or left foot. Levene’s test was performed to test for homogeneity of variance, which was present for all variables in each group, except for the area in bipedal stance with open eyes (Area_O). To identify significant differences between groups (*p* < 0.05), parametric and nonparametric tests were applied: Student’s t test for normally distributed data (Table 1) and the Mann-Whitney U test for all linear and nonlinear measures (Table 2). Considering the small sample size and nonnormal data, Spearman correlation analysis was used to find relationships concerning linear and nonlinear measures.

## 3. Results

Anthropometric measurements, body composition and physical test data are presented in Table 1. The mean difference between groups was highly significant for height (5.62, *p* < 0.001), 6MWT (3.46, *p* = 0.004), height/foot length ratio (H/FL) (3.43, *p* = 0.004), and PAS (2.14, *p* < 0.05), with large effect sizes (height, d = 3.04; 6MWT, d = 1.87, H/FL d= 1.845 and PAS d = 1.15).

A total of thirty linear and nonlinear measures were analyzed, with six presenting significant differences between groups (*p* < 0.05), showing strong effect size, and mostly for the right foot, which was dominant in 14/16 participants. Considering the small group sizes (*n* = 5 and *n* = 11), the critical value for significance in a two-tailed Mann-Whitney U test, at α = 0.05, was 9.

The LL group consistently presented larger differences in means in all linear measures (ToTex, A, and area) while the N group presented larger means in all nonlinear measures (SaEn and CoDim), in all tasks: bipedal stance with eyes open and closed and unilateral stance on the left and right foot. Data are available in Appendix A, Table A1. Stabilograms enable a visual representation of postural displacements in the ML and AP directions, with ToTex tracings in the bilateral eyes-closed (A) and unilateral left foot (B) tasks being represented in Figure 1. The traces, collected over 30 s of quiet standing, show in both stabilograms an increased amplitude and area traversed by the LL group comparing to N. In the bilateral task, the mean Area_C traversed by the LL group was 34.13 cm higher while A_ML was 1.99 cm and A_AP was 1.54 cm higher than those of the N. In the unipedal task, LL presented even greater displacements and compared to N, a higher difference of 13.11 cm for traversed area, 6.19 for A_ML and 4.36 for A_AP.

Several correlations were found between anthropometric metrics, strength evaluations, and postural measures, as presented in Table 3. It was with HGS that highest number of correlations was found, either in bilateral or in unilateral tasks, where the most significant was the correlation with CoDim_ML_R (*p* < 0.001). Push-ups, physical activity score, and height-to-foot ratio (H/F) presented only one correlation each with linear measures. The nonlinear measures presented the most correlations with HGS and FL.

Significant relationships were also observed through regression models between HGS and nonlinear variables. The most significant regressions were in bipedal tasks with SaEn_AP_O (R^2^ = 0.390, *p* = 0.01), with the most dispersed data, and CoDim_AP_C (R^2^ = 0.494, *p* = 0.002). In unipedal positions, the regressions found were with CoDim_ML_L (R^2^ = 0.583, *p* = 0.002) and CoDim_ML_R (R^2^ = 0.625, *p* < 0.001), as shown in Figure 2.

## 4. Discussion

This study provides novel insights into postural control strategies in adults with achondroplasia, a scarcely researched population. Heterogeneity existed regarding prior limb lengthening status, making it possible to evaluate two distinct subgroups: those with natural development (N) and those who had undergone lower limb lengthening (LL).

Based on the dynamical systems theory [16,21,42], multiple domains were analyzed to capture a more comprehensive evaluation of static standing ability in this population. Previous research found that linear metrics plateaued at mild impairment, while nonlinear analysis still detected control degradation in early disease [19]. Considering that achondroplasia is a lifelong chronic condition with progressive physical decay, this study applied a dual analytical approach by pairing linear and nonlinear analysis. The linear measures total excursion, amplitude and area were chosen to quantify absolute CoP deviations and for insights into postural control processes. However, while linear measures allow to capture the overall magnitude of body sway and gross deficiencies stabilizing equilibrium, these disregard the rich temporal structure of signals [21]. Therefore, nonlinear techniques were applied, as these better detect the intricacies of postural regulation, revealing subtleties in control processes from a moment-to-moment model, outlining the complex and erratic nature of postural corrections over time, unveiling the complexity of sway in this specific population.

Several characteristics emerged in the overall sample for this study on sway patterns. Having achondroplasia imposes challenges in the sensorimotor integration underlying balance regulation [33,43]. While the N group also incurred postural constraints from disproportionate limbs, the intervention effect inflicted further control difficulties, clearly highlighted by the LL group. LL showed higher means for all linear measures and lower means for nonlinear ones. The LL group exhibited amplified linear sway indicative of larger deviations, which may stem from lengthening surgery overstretching soft tissues, impairing proprioceptive feedback essential for balance [44]. On the other side, the N group exhibited more erratic and unpredictable sway, which has been observed in a healthy adaptive capacity [21], with a more fluid adaptation to positional perturbations. Although no other studies have been conducted on proprioception after limb lengthening surgery, studies in an older adult population observed an increased absolute angular error in hip and knee joints and impaired proprioception after hip fracture [45], which, in a certain way, by extrapolation to a surgical fracture as performed in LL, may cause loss of sensory information or have proprioceptive degradation posed by the intervention.

The lower SaEn and CoDim shown by the LL group together indicate relative adoption of more rigid and repeated and less complex postural control strategies compared to the N group. While lacking healthy controls precludes claims of absolute impairment or normality in either group, comparing nonlinear metrics between the LL and N groups provides relative indications of altered balance strategies and deficiencies in sensory–biomechanical adaptation from the limb lengthening intervention versus natural developmental constraints.

This suggests the intervention may hinder dynamic flexible adjustments as well as more prompt spatial correction essential for balance regulation, as highlighted in previous research [21,37,44]. The reduced adaptability and dimensionality can potentially stem from impairment of sensorimotor integration mechanisms under the altered skeletal morphology imposed by limb lengthening surgery over natural growth and development constraints. In contrast, the relatively higher, more erratic SaEn and multidimensional CoDim among the N group implies retention of greater postural variability and intricacy characteristic of systems still possessing some capacity to adapt control schemes moment-to-moment and in response to perturbations in space. These nonlinear dynamics spotlight likely further degradation versus preservation of residual neuromuscular networks integral to posture regulation following profound anatomical proportion changes, which can also be related to the amount of lengthening [46], with a length increase of over 20 cm being considered extensive lengthening [9]. Bone healing after intervention varies widely, from 6 months up to 2 years [47], whereas the time elapsed after intervention in the LL group varied between 5 and 32 years. Beyond limb lengthening being an option for AwACH to increase longitudinal height and surgical success, understanding postural control before and after this intervention is nearly nonexistent.

The moderate to strong positive correlations found between physical conditioning variables, in particular handgrip strength, in bilateral and unilateral tasks in linear and nonlinear variables, could be used as a first observation of balance status. These preliminary findings can also add on benefits for physical activity and targeted training in this population, which could also benefit functionality [48,49]. Muscle strength helps stabilize posture, and HGS correlations also highlight the vital role of muscle capacity in posture regulation for adults with achondroplasia. However, as the sample size of this study is limited, further research is critical to confirm the observed tendencies. Additionally, the relationships with nonlinear dynamics are a source of insight, as this makes it possible to observe the lack of variability and adaptability that characterizes improper control. The height-to-foot-length ratio correlated with sway area, implying larger oscillations with greater lower limb elongation relative to foot support. Interestingly, several correlations were found between foot length and linear and nonlinear measures, which have previously being recognized as factors in gait pattern alterations [50]. Though LL increased height, the unchanged smaller foot size compounded instability from shifted weight distribution, which was not overcome with time after intervention. The results presented in this study may present added points to be included in the decision-making process for future interventions. As knowledge expands regarding on balance constraints after intervention, personalized tailored rehabilitation, and training programs, can be implemented to offset deficiencies. The positive correlations of sway metrics with IPAQ physical activity scores imply activity participation is associated with balance regulation in achondroplasia. This aligns with previous research showing that exercise preserved mobility in aging through measurable balance impacts [29]. Critically, reduced activity levels often stem from a lack of confidence in facing motor challenges. While achondroplasia and surgery impose lifelong postural constraints, targeted training may enhance the capacity to withstand perturbations, which is critical for limiting avoidance behaviors and promoting participation.

### Strengths and Limitations

To our knowledge, this is the first study on postural control in adults with achondroplasia who underwent lower limb lengthening surgery. Recruitment was highly challenging, as achondroplasia is a rare condition, and a limited number of participants were available. Results must be read carefully, as the study included only 16 participants, and groups may not be representative of the AwACH subpopulations. The LL group was smaller than N group, and participants presented a wide range of time after the limb lengthening surgery as well as in the total gain in height. These variables may have critical impacts in postural control analysis. While the lack of healthy controls precludes claims of absolute impairment or normality in either group, the comparative nonlinear analysis still provides relative indications of differences between the LL and N groups regarding the impact of intervention on posture regulation dynamics.

## 5. Conclusions

This study provides novel insights into postural control strategies adopted by adults with achondroplasia, spotlighting distinct balance-regulation patterns influenced by natural development versus surgical limb lengthening. This study unveils insights into diversified postural control strategies adopted by adults with achondroplasia by spotlighting various factors integral to equilibrium regulation that are likely impaired in this population. Multiple domains of analysis quantified sway magnitudes and moment-to-moment control details, exposing signals of underlying sensory–biomechanical disruption. Having achondroplasia imposes lifelong postural constraints from disproportionate anatomy, further impacted by limb lengthening intervention. The limb lengthening group exhibited amplified yet more rigid sway, revealing overcorrections in alignment and proprioception that hinder dynamic reactions. In contrast, individuals with natural growth showed retention of the greater variability and unpredictability characteristic of systems that still possess some capacity to adapt.

Handgrip strength exhibited consistent correlations with posture metrics across tasks, underscoring its value as a potential marker for tracking functional instability and decline and further health conditioning parameters. Relationships with physical activity measures also emerged, highlighting movement’s influences on stability through neuromuscular adaptations. Foot length and height-to-foot-length ratio provided distinct cues on anatomical proportions influencing balance. Adults who had undergone limb lengthening demonstrated amplified yet more rigid sway patterns, with the need for overcorrections to alignment and proprioception that hinder dynamic equilibrium reactions. In contrast, individuals with natural growth exhibited more adaptive variability and unpredictability, characteristic of healthy systems.

It is estimated that around 250.000 PwACH exist worldwide [7], a reduced number of people compared with more prevalent conditions. However, as knowledge expands, stability optimization through individualized strength and activity promotion will become more likely and may afford this population the equitable opportunity to fulfil societal roles and activities that enrich wellbeing. Further research is needed to study rare populations with limited research attention, with the goal of achieving better care, enhanced social life, and physical activity inclusion.

## Figures and Tables

**Figure 1 jfmk-09-00039-f001:**
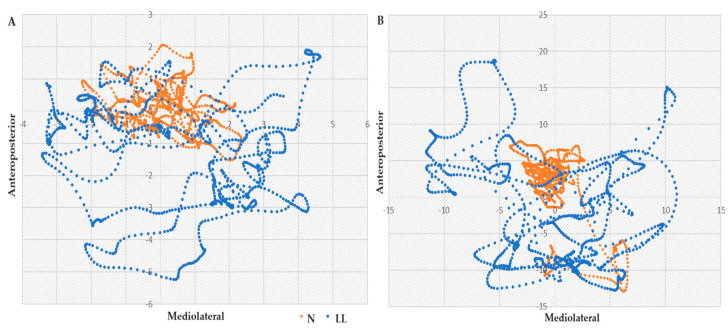
Stabilograms of the total excursion in the ML and AP sway in (**A**) bipedal eyes closed and (**B**) unipedal left tasks, for N (orange) and LL (blue) groups. Displacements presented in cm.

**Figure 2 jfmk-09-00039-f002:**
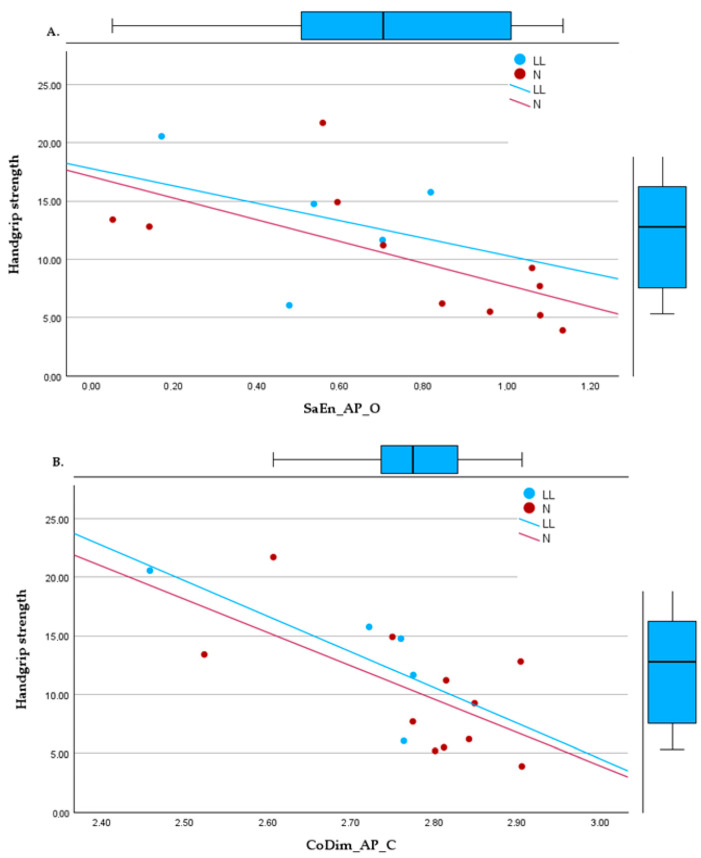
Linear regressions between handgrip strength and (**A**) sample entropy, bipedal, eyes open, in the anteroposterior direction; (**B**) correlation dimension, bipedal, eyes closed, in the anteroposterior direction; (**C**) correlation dimension in the mediolateral direction, right foot support; (**D**) correlation dimension in the mediolateral direction, left foot support.

**Table 1 jfmk-09-00039-t001:** Participants’ demographic, anthropometric, and physical activity data. Values are reported as the mean and standard deviation.

Data	Measurement	Group N (*n* = 11)	Group LL (*n* = 5)
Gender (F, M)	Female/male	7 F + 4 M	3 F + 2 M
Age	Years	40.50 ± 14.40	30.00 ± 8.92
Weight	kg	52.10 ± 16.70	57.30 ± 8.12
Height	cm	118.00 ± 7.53	141.00 ± 6.83
Foot length (FL)	cm	19.80 ± 1.72	20.60 ± 1.24
Height/FL (H/F)	ratio	6.00 ± 0.45	6.85 ± 0.49
Fat mass (FM)	Percentage	29.90 ± 10.10	21.60 ± 9.74
Handgrip (HGS)	kg	10.20 ± 5.31	13.80 ± 5.36
Push-ups (PU)	Count	16.60 ± 6.65	16.40 ± 13.40
6MWT	Meters	359.00 ± 69.50	479.00 ± 50.60
PAS	Total MET min/week	559.40 ± 725.10	1408.40 ± 761.10
PAL	Physical activity level	1 ± 1.00	2 ± 2.00

**Table 2 jfmk-09-00039-t002:** Linear and nonlinear measures with significant differences between groups.

Measure	U Statistic	Mean Difference	Effect Size (r_rb_)
ToTex_AP_R	9.00	306.41	0.673
A_ML_C	7.00	17.99	0.745
A_AP_R	7.00	35.27	0.745
Area_O	6.00	247.68	0.782
SaEn_AP_C	9.00	−0.33	0.673
CoDim_ML_R	8.00	−0.11	0.709

Abbreviations: r_rb_, rank-biserial correlation coefficient.

**Table 3 jfmk-09-00039-t003:** Significant correlations, presented as Spearman coefficients (rho), in bipedal and unipedal tasks. Significance levels of *p* < 0.05 and *p* < 0.01 (*), *p* < 0.001 (^†^).

	Weight	Height	HGS	PU	6MWT	PAS	FL	H/FL
ToTex_L	0.638 *		0.629				0.546	
ToTex_R								
ToTex_ML_L	0.618		0.656 *				0.515	
ToTex_ML_R							0.512	
ToTex_AP_L	0.691 *	0.519	0.735 *				0.705 *	
ToTex_AP_R								
Area_C								
Area_O		0.578			0.661 *			0.562
Area_R						0.514		
A_ML_C					0.518			
A_ML_L			0.574	0.629			0.512	
A_ML_R								
A_AP_L	0.559		0.697 *				0.685 *	
A_AP_R			0.550				0.518	
SaEn_AP_C			−0.526					
SaEn_AP_O			−0.650 *				−0.558	
SaEn_ML_R			−0.517					
SaEn_AP_L	−0.574		−0.665 *					
SaEn_AP_R			−0.538 *				−0.515	
CoDim_AP_C			−0.715 *					
CoDim_ML_O			−0.515				−0.507	
CoDim_ML_L	−0.653 *		−0.741 *					
CoDim_ML_R		−0.597	−0.815 ^†^		−0.579		−0.509	
CoDim_AP_R			−0.609					

## Data Availability

The data presented in this study are openly available in the Open Science Framework repository at https://osf.io/ype4a (accessed on 22 December 2023).

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
