# Peer review of "Unveiling the Chaos in Postural Control in Adults with Achondroplasia"

_jfmk, 2024, doi:10.3390/jfmk9010039_

Round 1

Reviewer 1 Report

Comments and Suggestions for Authors

The goal of this paper was to examine postural control in a group of individuals with achondroplasia with natural growth versus a group who had leg lengthening surgery.  Overall the study was well-designed. However, the presentation of the results is challenging to understand.  Furthermore, the rarity of the study population limits its significance and scientific interest.   Below are some specific suggestions for improvement. 

Line 45: please add incidence of limb lengthening in this population

Table 1: There is a large difference between total MET minutes in these two groups but this is not discussed, better to move this table to results where data are discussed.  

Figure 1 is not helpful, could remove since it was only a sample of two individuals with 1 trial. 

Table 3: it would be helpful to see some of these more significant correlations visually in graphic form. 

Figure 2: The violin plots are very challenging to understand.  Justification and greater explanation for these plots is needed, would be more powerful if they were typical column plots.  

line 263-264: there is no table or graph to show this relationship. 

line 270-273- Please discuss the significance of these results to the population or clinical care. How is it going to change treatment- therapy for these people.   

line 317-318: overstatement of results due to small sample and moderate correlation. 

line 342-343: important statement about small incidence, please report and discuss why it is of interest to study this population (see comment above). 

Comments on the Quality of English Language

There are several places in the manuscript where editing for spelling and grammar are needed

Line 44- change 'designated' to 'defined'

line 45-46- rewrite, improper grammar

line 63-change to "human posture is regulated via several feedback loops..."

line 78-79- rewrite needed

line 97-100- rewrite needed, awkward grammar

line 97- remove also

line 101-103- separate into new sentences, very long 

line 127- don't start sentence with 'but', need a space between while and muscle

line 196-197- awkward wording, needs rewrite. 

Reviewer 2 Report

Comments and Suggestions for Authors

This manuscript investigated characteristics in postural control in adults with Achondroplasia with natural growth and with lower limb lengthening using linear and non-linear measures. The main findings of this study were that the participants with lower limb lengthening showed significantly higher values of linear measures and lower values of non-linear measures such as sample entropy compared to ones with natural growth. Furthermore, the correlation between the measure of postural control and hand grip strength, and physical activity score. According to these results, the authors suggest that individuals with lower limb lengthening show more postural impairments and rigid sway patterns compared to ones with natural growth. In addition, the results indicate that individuals with Achondroplasia compensate for their postural impairments with muscle strength.

My major concerns are as follows:

1.         Methods: Although many demographic data are shown in Table 1, were these measurements collected on the same day as the posture control task? Also, how were these measurements collected, especially the PU?

2.         Table 1: What does “*” mean?

3.         Statistical analysis: It is unclear which statistical analysis method was used for what purpose. Please mention more details. In particular, there was no description for correlation analysis.

4.         Figure 1: the representative stabilogram of participants is shown in Figure 1. However, there was no description of the characteristics of the data. Please add a few sentences about what the authors want to tell the readers using this figure.

5.         Table 2: What does the “Statistic” mean? Also, I feel not only effect size but also the raw data of both groups are needed in the table.

6.         Table 3: I recommend that “*” be removed from the table. The marks would make it harder to understand the results.

7.         Figure 2: I believe that the legend is not position but task. Also, Abbreviations should not be used for the legend.

8.         Discussion: References are missing in many places. For example, lines 274, 291, and 331.

9.         Line 290: The authors mentioned that the lower limb lengthening surgery reduces sensory inputs, which contributes the postural impairments. Has sensory input decreased over time since surgery? Please explain it with appropriate references.

10.     The authors mentioned that the non-linear measurements have two sides. For example, less sample entropy indicates rigidity, while higher one indicates noisy postural control and unstable. Therefore, the results in this study cannot mention the subjects with lower limb lengthening have postural impairments. I believe that a comparison with healthy controls is necessary to show it. So, this would be a limitation.

11.     Discussion: The subjects with lower limb lengthening showed a wide range of duration after the surgery and total gain in height. I believe that these variations would affect postural control. Are there any effects in postural control measurements? Also, this would be a limitation.

Round 2

Reviewer 1 Report

Comments and Suggestions for Authors

Thank you for your detailed response to the reviewer comments.  These changes have enhanced the paper significantly.  I am in favor of publication.  

Reviewer 2 Report

Comments and Suggestions for Authors

The authors have sincerely attended to my comments and concerns. I appreciate your thorough scrutiny of the concerns raised by the reviewer. However, I have to comment on a point.

Point 10: I know you try to investigate the differences between people with limb lengthening and not, and then, you compare people with limb lengthening and not, but not healthy controls. If so, on what basis did you determine the results of the nonlinear analysis? Did the LL group show mean that was too low? Is it not just approaching normal?

Round 3

Reviewer 2 Report

Comments and Suggestions for Authors

The authors have efficiently attended to respond to my comment. I appreciate your thorough scrutiny of the concerns raised by the reviewer.